# Symmetry-preserving graph attention network to solve routing problems at multiple resolutions

## Abstract

Travelling Salesperson Problems (TSPs) and Vehicle Routing Problems (VRPs) have achieved reasonable improvement in accuracy and computation time with the adaptation of Machine Learning (ML) methods. However, none of the previous works completely respects the symmetries arising from TSPs and VRPs including rotation, translation, permutation, and scaling. In this work, we introduce the first-ever completely equivariant model and training to solve combinatorial problems. Furthermore, it is essential to capture the multiscale structure (i.e. from local to global information) of the input graph, especially for the cases of large and long-range graphs, while previous methods are limited to extracting only local information that can lead to a local or sub-optimal solution. To tackle the above limitation, we propose a **m**ultiresolution scheme in combination with **E**quivariant **G**raph **A**ttention network (mEGAT) architecture, which can learn the optimal route based on low-level and high-level graph resolutions in an efficient way. In particular, our approach constructs a hierarchy of coarse-graining graphs from the input graph, in which we try to solve the routing problems on simple low-level graphs first, then utilize that knowledge for the more complex high-level graphs. Experimentally, we have shown that our model outperforms existing baselines and proved that symmetry preservation and multiresolution are important recipes for solving combinatorial problems in a data-driven manner. Our source code is publicly available at **[anonymous url]**.

## 1 Introduction

Routing problems, such as TSPs and VRPs, are a class of NP-hard combinatorial optimization (CO) problems with numerous practical and industrial applications in logistics, transportation, and supply chain (Feillet et al., 2005; Laporte, 2009). Consequently, they have attracted intensive research efforts in computer science and operations research (OR), with many exact and heuristic approaches proposed (Croes, 1958; Applegate et al., 2009; Helsgaun, 2017; Gurobi Optimization, 2021). Exact methods often suffer from the drawback of high time complexity, making them impractical for real-world applications where instances have large sizes. Heuristic methods are more commonly used in practical applications, providing near-optimal solutions with much lower computational time and complexity. However, these algorithms lack guarantees of optimality and non-trivial theoretical analysis and heavily rely on handcrafted rules and domain knowledge, leading to difficulty in generalization to other combinatorial optimization problems. Therefore, it is important to build efficient data-driven methods that can learn to solve these challenging problems.

In recent times, machine learning methods (especially deep and reinforcement learning) inspired by Pointer Network (Vinyals et al., 2015), Graph Neural Network (Wu et al., 2020), and Transformer (Vaswani et al., 2017) have been used to automatically learn complex patterns, heuristics, or policies from generated instances for both TSPs and VRPs. These approaches are collectively referred to as *learning-based* (or data-driven) methods and are promising alternatives that can reduce computation time while maintaining solution quality compared to traditional *non-learning-based* methods. However, these machine learning models still face two major challenges: *Firstly*, since routing problems, e.g., TSP and VRP, are often represented in the 2D Euclidean space, their solutions must be invariant to geometric transformations of the input city coordinates, such as rotation, permutation, translation,

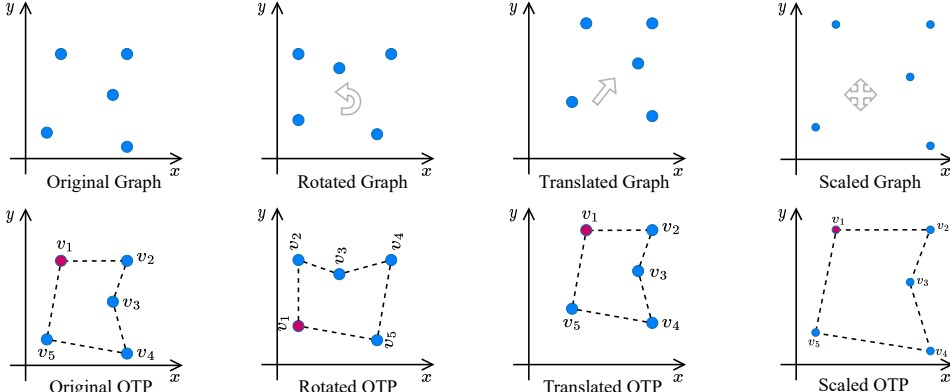

Figure 1: Invariant TSP with node coordinate transformations: rotation, translation, and scaling. Note that when we apply any of these geometric transformations to the original graph (in the first row), the optimal solution (OTP) of the instance remains unchanged (in the second row). This is one of the necessary invariant properties that deep models need to respect for routing problems.

and scaling (see Figure 1). Nevertheless, most existing *sequence-to-sequence* machine learning-based methods, e.g., Attention Model (Kool et al., 2019), have not yet fully considered the underlying symmetries of routing problems. *Secondly*, current learning-based methods often focus on learning from local information (spatial locality) to improve solution quality, which may lead to the model converging to local optima and missing leveraging global information.

In this paper, we aim to incorporate a geometric deep learning architecture into our model to enforce invariance (i.e. symmetry preservation) with respect to rotations, reflections, and translations of the input city coordinates. More specifically, we present an Equivariant Attention Model with Multiresolution architecture training for guiding the model to learn at multiple resolutions and multiple scales while remaining invariant to geometric transformations. To learn information from multiple resolutions and scales, we divide a problem instance into smaller sub-instances, each with its own feature, such as proximity space. We then build higher-level instances based on these sub-instances at a lower level and synthesize their information. These multiresolution instances are fed into the model to train together with the original instance using shared network weights to help the model generalize better and converge faster. Accordingly, our main contributions are summarized as follows:

- We first identify the equivariance and symmetries in deep (reinforce) models for solving routing problems and suggest incorporating an Equivariant Graph Attention model to respect invariant transformations on the input graphs. Such equivariant models are more stable than existing non-equivariant models when dealing with CO problems with invariant transformations, e.g., TSPs and VRPs.

- We propose *Multiresolution graph training* for learning routing problems at multiple levels, i.e. sub-graphs and high-level graphs. Our model is derived from Attention Model with Equivariant components and trained with sharing weights of different scales of an original problem instance to quickly capture both local and implicitly global structures of problems.

- We validate the efficiency and efficacy of our model on synthetic and real-world large-scale datasets of TSP with various distributions, sizes, and symmetries. We also devise our model for another routing problem, e.g., Capacitated Vehicle Routing Problem (CVRP). The obtained results show our model could have better generalization and stability compared to previous data-driven methods based on Attention Model in terms of solution quality.

## 2 RELATED WORK

**Classical methods.** Classical methods for TSPs and VRPs have been extensively studied in the literature, typically including exact and approximate (approximation, heuristic) approaches (Laporte, 1992a;b). For TSPs, exact algorithms such as branch and bound (Clausen, 1999) and dynamic

programming (Bellman, 1962) and heuristic methods such as the nearest neighbor (Held & Karp, 1962), 2-opt (Croes, 1958), and LKH (Lin & Kernighan, 1973) are well-known. Similarly, for VRPs, classical methods include exact algorithms such as branch and cut (Lysgaard et al., 2004), and heuristics such as Tabu search (Glover & Laguna, 1998), Genetic Algorithms (Holland, 1992), and LKH-3 (Helsgaun, 2017). Although these methods have successfully solved small to medium-sized instances of TSP and VRP, they often struggle to scale up to large-scale problems due to their exponential growth in time complexity.

**Neural methods.** In CO, deep learning methods have been applied and obtained promising results and better scalability. The popular deep architectures are comprised of two components, i.e. encoder and decoder (Sutskever et al., 2014; Kool et al., 2019; Joshi et al., 2019). Recent researches tend to focus more on building efficient decoders and loss functions. Kool et al. (2019) generates solutions in an autoregressive manner, whereas Joshi et al. (2019) generates solutions directly. Besides, constructing labels for large-scale problems is often costly or infeasible; hence using reinforcement learning to train models is more appropriate and attractive (Peng et al., 2020; Kwon et al., 2020). Cheng et al. (2023); Li et al. (2021); Xin et al. (2021) try to improve the results by finding and re-optimizing sub-paths, then merging them into the original solution via a heuristic way. Some efforts have been made to address the issue of symmetries for TSPs, such as POMO (Kwon et al., 2020) and Sym-NCO Kim et al. (2022). However, these methods are not fully optimized and cannot respect geometric transformations. Different from previous studies, our work focuses on building an equivariant deep model that can systematically extract problem information in the encoder at multiple levels, which can jointly help the decoder generate a better solution and preserve invariance to geometric transformations.

**Symmetry preserving methods.** Starting with (Cohen & Welling, 2016), group equivariance and symmetry preserving have emerged as core organizing principle of deep neural network architectures. From classical Convolutional Neural Networks (CNNs) (LeCun et al., 1989; Krizhevsky et al., 2012) that are equivariant to translations, researchers have also constructed neural networks that are equivariant to the Euclidean group of translations and rotations (Cohen & Welling, 2017; Weiler et al., 2018), the three dimensional rotation group (Cohen et al., 2018; Anderson et al., 2019), the permutation group in the context of learning on sets (Zaheer et al., 2017) and learning on graphs (Maron et al., 2019; Kondor et al., 2018; Hy et al., 2018; 2019), and other symmetry groups (Ravanbakhsh et al., 2017). Based on the existing development of equivariant neural networks, we propose a general learning architecture for solving combinatorial problems (e.g., TSPs and VRPs) that respects all underlying symmetries including rotation, permutation, translation, and scaling.

**Multiresolution learning on graphs.** Graph neural networks (GNNs) that generalize the concept of convolution to graphs (Scarselli et al., 2009; Niepert et al., 2016) have been widely applied to several applications including modeling physical systems (Battaglia et al., 2016), predicting molecular properties (Kearnes et al., 2016; Gilmer et al., 2017; Hy et al., 2018), learning to solve NP-complete (Prates et al., 2019) and NP-hard problems (Dai et al., 2017), etc. Most GNNs are constructed based on the message passing scheme (Gilmer et al., 2017) in which each node propagates and aggregates to and from its local neighborhood. Because of the restriction to locality, the message passing scheme lacks the ability to capture the multiscale and hierarchical structures that are present in complex and long-range graphs (Dwivedi et al., 2022). (Hy & Kondor, 2023; Ngo et al., 2023) proposed a learning to cluster algorithm in a data-driven manner that iteratively partitions a graph and coarsens it in order to build multiple resolutions of a graph, i.e. multiresolution. In our work, we adopt the multiresolution framework to capture both local and global information of the input structure that are essential in solving combinatorial problems.

## 3 PRELIMINARIES

### 3.1 ROUTING PROBLEM VIA MARKOV DECISION PROCESS

We investigate the routing problems in 2D Euclidean space, where a problem instance $s$ is defined as a graph $G = (V, E)$ with a set of $n$ nodes $V = \{v_1, v_2, ..., v_n\}$. Each node $v_i$ is represented by a 2D coordinate. The edges set $E$ represent connections between these nodes, and the edges are weighted by the Euclidean distance. In many cases, a solution to the problems can be represented as a sequence of nodes. The optimization objective of these problems is commonly to minimize the total traveling distance (length). For a simple illustration, we take a TSP as a representative example,

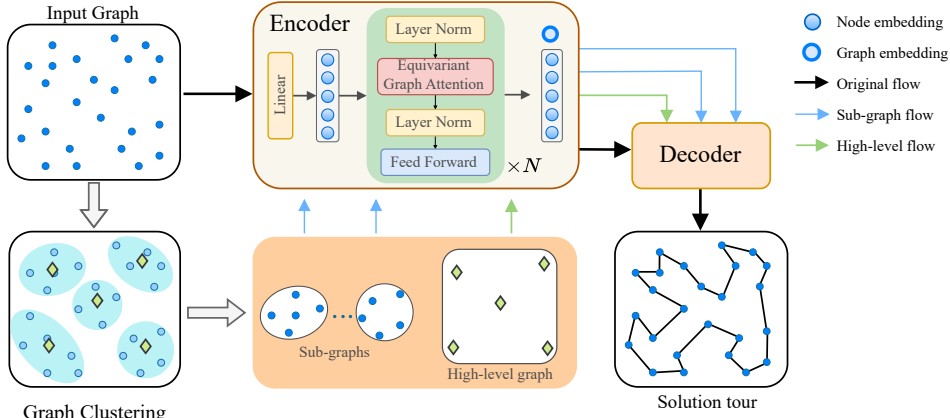

Figure 2: **Overview of our mEGAT framework**. Given an input graph, our multiresolution graph construction first output all sub-graphs and multiresolution high-level graphs, e.g., a bi-level graph, based on graph clustering. Then, we train our model at multi-level graphs by feeding the original graph, sub-graphs, and high-level graph into our model to jointly learn and optimize.

although our proposed approach can be applied to other routing problems, such as VRPs. A feasible solution for a TSP with $n$ cities, typically denoted TSP$n$, is defined as a *tour* $\boldsymbol{\pi} = (\pi_1, ..., \pi_n)$ as a permutation of the $n$ nodes that each node is visited once and only once. The length of a tour is defined as: $\mathbb{L}(\boldsymbol{\pi}) = \|c_{\pi_n} - c_{\pi_1}\| + \sum_{i=1}^{n-1} \|c_{\pi_i} - c_{\pi_{i+1}}\|_2$, where $c_i$ is the coordinate of node $v_i$ and $\|.\|_2$ denotes the $l_2$ norm.

## 3.2 SYMMETRIES, GROUPS AND EQUIVARIANCE

In group theory, the collection of symmetries form an algebraic object known as a group. Formally, a group is defined by 4 axioms (i.e. closure, associativity, identity and inverse) as in Def. A.1. In this work, we are particularly interested in the symmetry of permutation, rotation, translation, and scale. Our purpose is to design an equivariant model with respect to all these groups:

- A permutation of order $n$ is a bijective map $\sigma : \{1, 2, \cdots, n\} \to \{1, 2, \cdots, n\}$. The permutation group or symmetric group of degree n, denoted as $\mathbb{S}_n$, is the set of all $n!$ permutations of order $n$. Any graph neural network must be permutation-invariant or $\mathbb{S}_n$-invariant, i.e. regardless of how we permute the set of vertices, the output of GNN must remain the same because permutation does not change the graph topology. In TSP of $n$ cities, a permutation will reindex/rename the cities but never change the length of the optimal tour. We say that the (unique) optimal tour is $\mathbb{S}_n$-equivariant and its length is $\mathbb{S}_n$-invariant.
- The 2D or 3D rotation group or special orthogonal group, denoted as SO(2) or SO(3) respectively, is the group of all rotations about the origin of the Euclidean space under the operation of composition.
- The set of all translations $T_{\Delta x, \Delta y} : (x, y) \mapsto (x + \Delta x, y + \Delta y)$ form the translation group.

In TSPs and VRPs, we assume that locations are mapped into a 2D Euclidean space; and regardless of how we rotate, translate or scale (i.e. multiplying all the coordinates with a non-zero constant) them, the optimal solution remains unchanged (see Fig. 1). Any model trying to solve these combinatorial problems must be rotation-invariant or SO(2)-invariant, translation-invariant and scale-invariant. In summary, any solver must be $G$-invariant where $G$ is a group of any symmetry that arises from the input domain.

## 4 METHODOLOGY

**Motivation.** For complex TSPs or VRPs, an effective approach commonly employed in prior works is to divide an original problem into simplified sub-problems, solve them individually, and then combine

their solutions to form the final solution for the original problem. Our work is also motivated by this idea, but it differs from previous works in that we propose a framework capable of constructing both sub-problems and higher-level problems (see 4.1). Furthermore, we utilize these constructed problems in joint training with the original problem using our Multiresolution graph training (see 4.2). Additionally, we handle a symmetry-preserving graph attention network that can learn solutions invariant to geometric transformations (see 4.3). The overall approach is illustrated in Figure 2.

## 4.1 LEARNING MULTIPLE RESOLUTION GRAPHS

**Sub-graph construction.** As mentioned in 3.1, a routing problem in 2D Euclidean space is typically defined over an undirected graph $G = (V, E)$ with node set $V = \{v_i = (x_{v_i}, y_{v_i}) | x_{v_i}, y_{v_i} \in \mathbb{R}, i \in [[1, n]]\}$ and edge set $E = \{e = (v_i, v_j) | v_i, v_j \in V, i \neq j)\}$. We give some definitions as follows.

**Definition 4.1.** A $K$-subgraph construction of graph $G$ is a partition of the set of nodes $V$ into $K$ mutually exclusive cluster $V_1^{sub}, ..., V_K^{sub}$. Each cluster corresponds to an induced sub-graph $G_k^{sub} = G[V_k^{sub}] = (V_k^{sub}, E_k^{sub})$ in which $V = V_1^{sub} \cup \cdots \cup V_K^{sub}$ and $E_k^{sub} = \{e = (u, v) \in E | u, v \in V_k^{sub}\}$, $k \in [[1, K]]$.

**Definition 4.2.** Given a sub-graph construction, a high-level (i.e. coarsened graph) of a weighted graph $G = (V, E)$ is a weighted complete graph $G' = (V', E')$ which includes $K$ nodes corresponding to the $K$ clusters that are determined from the sub-graph construction. Each node $v'_k \in V'$ can be determined by a 2D Euclidean coordinate as:

$$x_{v'_k} = \frac{1}{|V_k^{sub}|} \sum_{v \in V_k^{sub}} x_v, \qquad y_{v'_k} = \frac{1}{|V_k^{sub}|} \sum_{v \in V_k^{sub}} y_v. \tag{1}$$

The weight of each edge $e' = (v'_i, v'_j) \in E'$ connecting $V_i^{sub}$ and $V_j^{sub}$ is determined as:

$$w(e') = d(v'_i, v'_j) = \|v'_i - v'_j\|_2, \tag{2}$$

where $d : \mathbb{R} \times \mathbb{R} \to \mathbb{R}$ denotes the Euclidean distance between two nodes.

**Definition 4.3.** A $L$-level multiresolution (i.e. multiple of resolutions) of graph $G$ is a series of $L$ graphs $G_1, \ldots, G_L$ where:

- $G_L$ is $G$ itself.
- For $1 \leq \ell \leq L - 1$, the $\ell$-th level graph $G_\ell$ is a high-level graph of $G_{\ell+1}$, as defined in Def. 4.2, where the number of nodes in $G_\ell$ is the number of clusters in $G_{\ell+1}$.
- The top-level $G_1$ is a graph with at least 5 nodes[1].

Algorithm 1 describes a simplified implementation of our multiresolution graphs construction with 2-level (i.e., $L = 2$), given an input graph $G$. We can repeat the algorithm on the existing-level graph to obtain a higher-level resolution graph. These corresponding sub-graphs could capture the local information, while these high-level graphs could capture the global information in a long range of the original graph. We subsequently utilize the features from these sub-

---

**Algorithm 1: Multiresoltion graphs construction**

**Input:** Input graph $G = (V, E)$.
$\{V_k^{sub}\}_{k=1}^K \leftarrow$ Clustering$(G)$;
$V' \leftarrow \varnothing$ and $E' \leftarrow \varnothing$;
**for** $k \leftarrow 1..K$ **do**
  $E_k^{sub} = \{e = (u, v) | u, v \in V_k^{sub}\}$;
  $G_k^{sub} \leftarrow (V_k^{sub}, E_k^{sub})$;
  $V' \leftarrow V' \cup \{v'_k\}$ with $v'_k = (x_{v'_k}, y_{v'_k})$ updated by Eq. 1;
  $E' \leftarrow E' \cup \{e'\}$ with $w(e')$ updated by Eq. 2;

**Output:** Sub-graphs $\{G_k^{sub} = (V_k^{sub}, E_k^{sub})\}_{k=1}^K$ and high-level graph $G' = (V', E')$.

---

graphs to facilitate solving the original problem. Our motivation is rooted in the observation that many combinatorial optimization problems exhibit inherent spatial locality, whereby the entities within the problem are more strongly influenced by their neighboring entities than those far away.

## 4.2 MODEL TRAINING AT MULTIPLE LEVEL GRAPHS

Most previous learn-to-struct methods, such as AM Kool et al. (2019) and POMO Kwon et al. (2020), use RL to train a sequence-to-sequence policy for solving routing problems directly. The correspond-

---

[1]In our experiments for TSP and VRP, the top-level graph has at least 5 nodes.

ing stochastic policy $p(\boldsymbol{\pi}|s)$ for selecting a solution $\boldsymbol{\pi}$ which is factorized and parameterized by $\boldsymbol{\theta}$ as:

$$p_{\boldsymbol{\theta}}(\boldsymbol{\pi}|s) = \prod_{t=1}^{n} p_{\boldsymbol{\theta}}(\pi_t|s, \boldsymbol{\pi}_{1:t-1}). \tag{3}$$

From the probability distribution $p_{\theta}(\boldsymbol{\pi}|s)$, the model can obtain a solution $\boldsymbol{\pi}|s$ by sampling. Normally, the expectation of the tour length (cost) is used as the training loss: $\mathcal{L}(\boldsymbol{\theta}|s) = \mathbb{E}_{p_{\boldsymbol{\theta}}(\boldsymbol{\pi}|s)}\left[\mathbb{L}(\boldsymbol{\pi})\right]$ on the original instance $s$.

In this paper, we use the constructed sub-problems (section 4.1) in joint training with the original problem to leverage local and global features to learn the problem more effectively. Given a problem instance $s$, we obtain a set of $K$ sub-instances $s_k^{sub}$, ($k \in [\![1, K]\!]$) and a set of $L$ high-level instances $s_\ell^{high}$ ($\ell \in [\![1, L]\!]$) after performing a clustering method on the original instance. Note that in our work, the terms *(sub, high-level) instance* and *(sub, high-level) graph* are equivalent and can be used interchangeably.

Motivated by the idea of multiple level-graphs training, we define a new loss function to help our model learn and generalize better in both the original graph and different multi-scale graphs. Our model learns a policy $p_{\boldsymbol{\theta}}$ by minimizing the total loss function:

$$\mathcal{L}_{\text{total}} = \mathcal{L}_{\text{original instance}} + \mathcal{L}_{\text{sub-instances}} + \mathcal{L}_{\text{high-level instances}}, \tag{4}$$

where $\mathcal{L}_{\text{original instance}}$, $\mathcal{L}_{\text{sub-instances}}$, and $\mathcal{L}_{\text{high-level instances}}$ are REINFORCE loss that needs to be optimized for original, sub, and high-level instances, respectively. The terms of two additional loss functions for sub and high-level instances are defined as follows:

$$\mathcal{L}_{\text{sub-instances}}(\boldsymbol{\theta}|s^{sub}) = \frac{1}{K} \sum_{k=1}^{K} \mathbb{E}_{p_{\boldsymbol{\theta}}(\boldsymbol{\pi}_k^{sub}|s_k^{sub})}\left[\mathbb{L}(\boldsymbol{\pi}_k^{sub})\right], \tag{5}$$

$$\mathcal{L}_{\text{high-level instances}}(\boldsymbol{\theta}|s^{high}) = \frac{1}{L-1} \sum_{\ell=1}^{L-1} \mathbb{E}_{p_{\boldsymbol{\theta}}(\boldsymbol{\pi}_\ell^{high}|s_\ell^{high})}\left[\mathbb{L}(\boldsymbol{\pi}_\ell^{high})\right], \tag{6}$$

where $\boldsymbol{\pi}^{sub}$ and $\boldsymbol{\pi}^{high}$ are solutions on sub-instance $s^{sub}$ and high-level instance $s^{high}$, respectively (e.g., $\boldsymbol{\pi}_k^{sub}$ denotes the $k$-th sub-instance's solution on low-level, and $\boldsymbol{\pi}_\ell^{sub}$ denotes the $\ell$-th resolution's solution on the high-level instance).

Different from previous studies, such as AM Kool et al. (2019) and POMO Kwon et al. (2020), our loss function guides the model to learn information of the instance $s$ at multiple scales by combining both global information (at the original graph and high-level graph) and local information (at sub-graphs). These pieces of information can help our model converge faster and prevent it from converging to local optima. Moreover, training on multiple graph scales simultaneously helps improve the generalization ability of our pre-trained model when solving instances of various sizes. The loss $\mathcal{L}$ can be optimized by gradient descent using REINFORCE (Williams, 1992) gradient estimator with a deterministic greedy rollout baseline used in AM Kool et al. (2019) or shared baseline for gradients used in POMO Kwon et al. (2020). Note that the shared baseline of POMO training is a significant improvement of AM. Therefore, in this work, we use the same REINFORCE algorithm in POMO to train our model for better performance. More details of training and baseline policy can be found in Appendix B. Algorithm 2 presents our multiple resolutions graph training with minibatch in one epoch.

---

**Algorithm 2: Multisolution graph Training**

**Input:** steps per epoch $T$, batch size $B$, number of clusters $K$, number of resolution levels $L$.
Initialize policy network parameter $\boldsymbol{\theta}$;
**for** *step* $= 1, \ldots, T$ **do**
    $s_i \leftarrow$ SampleInstance() $\forall i \in \{1, \ldots, B\}$;
    $\{s_{i_k}^{sub}\}_{k=1}^{K}, \{s_{i_\ell}^{high}\}_{\ell=1}^{L} \leftarrow$ Multiresolution graphs construction according to Algorithm 1;
    $\boldsymbol{\pi}_i, \boldsymbol{\pi}_{i_k}^{sub}, \boldsymbol{\pi}_{i_\ell}^{high} \leftarrow$ SampleRollout($s_i, s_{i_k}^{sub}, s_{i_\ell}^{high}, p_{\boldsymbol{\theta}}$) $\forall i \in \{1, \ldots, B\}$;
    $\boldsymbol{\pi}_i^{\text{BL}}, \boldsymbol{\pi}_{i_k}^{sub\text{BL}}, \boldsymbol{\pi}_{i_\ell}^{high\text{BL}} \leftarrow$ GreedyRollout($s_i, s_{i_k}^{sub}, s_{i_\ell}^{high}, p_{\boldsymbol{\theta}}^{\text{BL}}$) $\forall i \in \{1, \ldots, B\}$;
    $\nabla\mathcal{L} \leftarrow \sum_{i=1}^{B} \left(\Delta_{\mathbb{L}}(\boldsymbol{\pi}_i) + \frac{1}{K}\sum_{k=1}^{K}\Delta_{\mathbb{L}}(\boldsymbol{\pi}_{i_k}^{sub}) + \frac{1}{L-1}\sum_{\ell=1}^{L-1}\Delta_{\mathbb{L}}(\boldsymbol{\pi}_{i_\ell}^{high})\right) \nabla_{\boldsymbol{\theta}} \log p_{\boldsymbol{\theta}}(\boldsymbol{\pi}_i)$, where
    $\Delta_{\mathbb{L}}(\boldsymbol{\pi}_x) = \mathbb{L}(\boldsymbol{\pi}_x) - \mathbb{L}(\boldsymbol{\pi}_x^{\text{BL}})$;
    $\boldsymbol{\theta} \leftarrow$ Adam($\boldsymbol{\theta}, \nabla\mathcal{L}$);

---

### 4.3 Equivariant Attention Model

We use an encoder-decoder architecture similar to the Attention Model (AM) (Kool et al., 2019). However, we replace the Encoder with an Equivariant Graph Encoder to respect the equivariant property with geometric transformations on the input graphs. The Decoder is used in the same manner as the architecture of AM.

**Equivariant Graph Attention Encoder.** Take the $d_{\text{x}}$-dimensional input features $\mathbf{x}_i$ of node $v_i$ (for TSP, $d_{\text{x}} = 2$), the encoder uses a learned linear projection with parameters $W^{\text{x}}$ and $\mathbf{b}^{\text{x}}$ to computes initial $d_{\text{h}}$-dimensional node embeddings $\mathbf{h}_i^{(0)}$ by $\mathbf{h}_i^{(0)} = W^{\text{x}}\mathbf{x}_i + \mathbf{b}^{\text{x}}$. After obtaining the embeddings, we feed them to $N$ equivariant attention layers. We denote with $\mathbf{h}_i^{(l)}$ the node embeddings produced by layer $l \in \{1, \ldots, N\}$. The encoder computes an aggregated embedding $\bar{\mathbf{h}}^{(N)}$ of the input graph as the mean of the final node embeddings $\mathbf{h}_i^{(N)}$: $\bar{\mathbf{h}}^{(N)} = \frac{1}{n} \sum_{i=1}^{n} \mathbf{h}_i^{(N)}$. Similar to AM, both the node embeddings $\mathbf{h}_i^{(N)}$ and the graph embedding $\bar{\mathbf{h}}^{(N)}$ are used as input to the decoder. For the attention mechanism, we use the Linear, Layer Norm, and Equivariant Graph Attention sublayers designed by Liao & Smidt (2022) that are equivariant operators. More details of these sublayers can be found in Appendix B.

**Decoder.** We use the same decoder as AM (Kool et al., 2019) that reconstructs the solution (tour $\boldsymbol{\pi}$) sequentially. Particularly, the decoder outputs the node $\pi_t$ at each time step $t \in \{1, .., n\}$ based on the context embedding coming from the feature embeddings of node and graph obtained by the encoder and the previous outputs $\pi'_t$ at timestep $t' < t$.

## 5 Experiments

### 5.1 Setup

**Baselines and Metrics.** We compare our proposed model to the solid baseline methods for both TSP and CVRP, including non-learnable baselines (heuristics): Concorde (Applegate et al., 2006), LKH3 (Helsgaun, 2017), and Gurobi (Gurobi Optimization, 2021) and recent neural constructive baselines: AM Kool et al. (2019), POMO Kwon et al. (2020), MDAM Xin et al. (2021), and Sym-NCO Kim et al. (2022). We use *tour length* (Obj), *optimality gap* (Gap), and *evaluation time* (Time) as the metrics to evaluate the performance of our models compared to other baselines. The smaller the metric values are, the better performance the model achieves.

**Datasets.** For *synthetic instances*, we follow the data generation in previous works (Kool et al., 2019; Kim et al., 2022) to generate 2D Euclidean instances where city coordinates are generated independently from a unit square $[0, 1]^2$ uniform distribution. We train our model using 20, 50, and 100-node instances. More details are shown in Appendix C. For *benchmark instances*, we use the test instances from real-world TSP problems with problem sizes from 50 up to 400.

**Training and hyperparameters.** To make fair comparisons, we use the same training-related hyperparameters from the original paper on neural baselines when applying our method to these models. We implement our models in Pytorch (Paszke et al., 2019) and train all models on a machine with server A100 40GPUs. For clustering, we perform the K-means algorithm and execute it in parallel on GPUs to speed up training time and select 2D coordinates of the cities as features, and we set $L = 2$ for our multiresolution graph construction. Please refer to Appendix C for more details.

### 5.2 Results on TSP and CVRP

We first report the testing results of our model for TSPs and VRPs compared with other baselines on the test instances with 20, 50, and 100 nodes in Table 1. The results are reported on an average of 10,000 test instances via *greedy* decoding, i.e., select the best action at each step, and *sampling*, i.e., sample 1280 samples and report the best. It can be seen that our models outperform the neural baselines for TSPs and CVRPs ($n$ =20, 50, and 100) in terms of solution quality and evaluation time. For the larger size in real-world TSP instances, we report the results in Table 3 (see Appendix D). The obtained results also demonstrate that our model performs better solutions in the fastest time compared with other neural baselines.

Table 1: Performance of our mEGAT and baseline for TSPs. 'g', 's', and 'bs' denote greedy, sampling, and the beam search method, respectively. The best results are in bold.

| | Model | n=20 Obj↓ | Gap | Time | n = 50 Obj↓ | Gap | Time | n = 100 Obj↓ | Gap | Time |
|---|---|---|---|---|---|---|---|---|---|---|
| **TSP** | Concorde | 3.84 | 0.00 % | 1m | 5.70 | 0.00 % | 2m | 7.76 | 0.00 % | 3m |
| | LKH3 | 3.84 | 0.00 % | 18s | 5.70 | 0.00 % | 5m | 7.76 | 0.00 % | 21m |
| | Gurobi | 3.84 | 0.00 % | 7s | 5.70 | 0.00 % | 2m | 7.76 | 0.00 % | 17m |
| | AM (g.) Kool et al. (2019) | 3.85 | 0.34 % | 0s | 5.80 | 1.76% | 2s | 8.12 | 4.53 % | 6s |
| | AM (s.) Kool et al. (2019) | 3.84 | 0.08 % | 5m | 5.73 | 0.52% | 24m | 7.94 | 2.26 % | 1h |
| | POMO (g.) Kwon et al. (2020) | 3.84 | 0.06 % | 0s | 5.75 | 0.71 % | 1s | 7.85 | 1.21 % | 3s |
| | POMO (s.) Kwon et al. (2020) | 3.84 | 0.02 % | 1s | 5.72 | 0.50 % | 3s | 7.80 | 0.48 % | 16s |
| | MDAM (g.) Xin et al. (2021) | 3.84 | 0.05 % | 5s | 5.73 | 0.62 % | 15s | 7.93 | 2.19 % | 45s |
| | MDAM (bs.) Xin et al. (2021) | 3.84 | 0.00 % | 5m | 5.70 | 0.03 % | 20m | 7.80 | 0.48 % | 1h |
| | Sym-NCO (g.) Kim et al. (2022) | – | – | – | – | – | – | 7.84 | 0.94 % | 2s |
| | Sym-NCO (s.) Kim et al. (2022) | – | – | – | – | – | – | 7.79 | 0.39 % | 16s |
| | **mGAT** (g.) (Ours) | 3.84 | 0.02 % | 1s | 5.73 | 0.62 % | 1s | 7.82 | 0.79 % | 3s |
| | **mGAT** (s.) (Ours) | **3.84** | **0.00** % | 3s | **5.70** | **0.02** % | 3s | **7.78** | **0.25** % | 16s |
| **CVRP** | LKH3 | 6.14 | 0.58 % | 2h | 10.38 | 0.00 % | 7h | 15.65 | 0.00 % | 13h |
| | Gurobi | 6.10 | 0.00 % | | | | – | | | – |
| | AM (g.) Kool et al. (2019) | 6.40 | 4.97 % | | 10.98 | 5.86 % | 3s | 16.80 | 7.34 % | 8s |
| | AM (s.) Kool et al. (2019) | 6.25 | 2.45 % | 9m | 10.62 | 2.40% | 28m | 16.23 | 3.72 % | 1h |
| | POMO (g.) Kwon et al. (2020) | 6.22 | 1.97 % | 2s | 10.54 | 1.56 % | 12s | 16.26 | 3.93 % | 3s |
| | POMO (s.) Kwon et al. (2020) | 6.19 | 1.48 % | 5s | 10.49 | 1.14 % | 6s | 15.90 | 1.67 % | 16s |
| | MDAM (g.) Xin et al. (2021) | 6.24 | 2.29 % | 15s | 10.74 | 3.47 % | 16s | 16.40 | 4.86 % | 45s |
| | MDAM (bs.) Xin et al. (2021) | 6.15 | 0.82 % | 6m | 10.50 | 1.18 % | 20m | 16.03 | 2.49 % | 1h |
| | SymNCO (g.) Kim et al. (2022) | – | – | – | – | – | – | 16.10 | 2.88 % | 3s |
| | Sym-NCO (s.) Kim et al. (2022) | – | – | – | – | – | – | 15.87 | 1.46 % | 16s |
| | **mGAT** (g.) (Ours) | 6.20 | 1.64 % | 2s | 10.50 | 1.15 % | 12s | 16.08 | 2.75 % | 3s |
| | **mGAT** (s.) (Ours) | **6.14** | **0.65** % | 5s | **10.46** | **0.77** % | 6s | **15.84** | **1.21** % | 16s |

## 5.3 GENERALIZATION STUDY

**Variable-size generalization.** We first test the generalization of our model trained on a larger size ($n = 100$) to various smaller sizes of 20 and 50. As shown in Table 2, the results indicate that our model trained on larger sizes could perform well on downgrade ones and better than all neural baselines such as AM, POMO, MDAM, and Sym-NCO. One possible reason is that our model (trained on TSP100) leverages joint training (multiresolution) on low-level graphs with smaller sizes. For scale-up sizes, the results on larger sizes (> 100) in TSPLib Reinelt (1991) in Table 3 also confirm the superior performance of our model compared to these previous attention-based models.

**Cross-distribution generalization.** We examine our model performance on the *cluster* and *mixed* distributions of TSP datasets. As shown in Table 4 in Appendix D, our model achieves better results for both distributions than POMO in all settings, especially in Mixed distribution, thus confirming the efficiency of our multiresolution learning from low-level and high-level graphs. Please see the Appendix D for more details.

Table 2: Performance of our model compared with baselines (all models trained on TSP100) on variable TSP sizes.

| | TSP20 Obj↓ | Gap | Time | TSP50 Obj↓ | Gap | Time | TSP100 Obj↓ | Gap | Time |
|---|---|---|---|---|---|---|---|---|---|
| AM Kool et al. (2019) | 3.97 | 3.39 % | 5m | 5.82 | 2.33 % | 14m | 7.94 | 2.26 % | 30m |
| POMO Kwon et al. (2020) | 3.89 | 1.06 % | 2s | 5.74 | 0.71 % | 6s | 7.80 | 0.48 % | 16s |
| MDAM Xin et al. (2021) | 3.95 | 2.86 % | 5m | 5.81 | 1.94 % | 28m | 7.80 | 0.48 % | 45m |
| Sym-NCO Kim et al. (2022) | 3.88 | 1.04 % | 3s | 5.73 | 0.53 % | 8s | 7.79 | 0.39 % | 17s |
| **mGAT** (Ours) | **3.85** | **0.27** % | 2s | **5.71** | **0.18** % | 6s | **7.78** | **0.25** % | 16s |

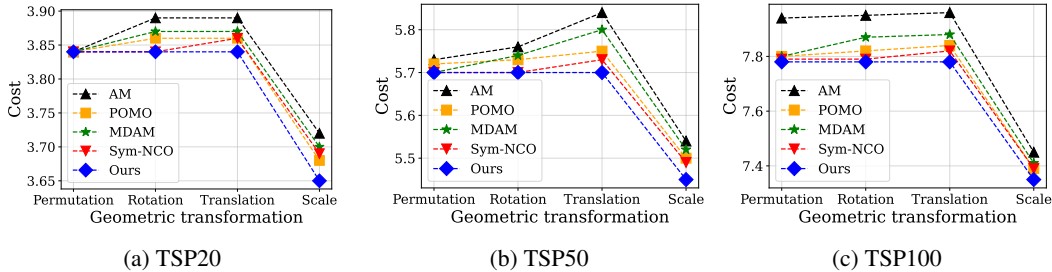

(a) TSP20       (b) TSP50       (c) TSP100

Figure 3: **Equivariance ablation analysis to geometric transformations.** (a) Performance on TSP20, (b) Performance on TSP50, (c) Performance on TSP100.

## 5.4 ABLATION STUDIES

**Equivariance analysis.** To validate the equivariant performance of our models on transformation data sets, we generate a new test set, including instances from the training set that undergoes three transformations: rotation, translation, and scaling. Particularly, we use a $90°$ counterclockwise rotation on the center point located at $[0.5, 0.5]$, a translation with mapping rule $(x, y) \mapsto (x + \Delta x, y + \Delta y)$ ($\Delta x, \Delta y \in [0, 1]$), and a random scaling of the distances in the range of $[0.7, 1.2]$. The results in Figure 3 show that our model is more stable than the baseline on the transformation datasets, thus confirming the crucial role of the equivariant deep model for CO problems.

**Multiple resolutions learning.** We conduct an ablation study on TSP50 and TSP100 to verify the design of our model, where we replace the multiresolution graph training (*w/o* multiresolution) with the same baseline training as POMO and equivariant graph attention encoder (*w/o* equivariant) with the same attention encoder as AM. As can be seen from Table 4, the use of multiresolution graph training further enhances the learning effectiveness, and the equivariant encoder plays a prominent role in the stability performance of our model.

Furthermore, we also evaluate the influence of the number of clusters $K$ when dividing the original graph into sub-graphs on the model's performance. We report the results with $K = 5$, $K = 10$ of our models trained on 100,000 samples on the fly and evaluated on the 10,000 test instances in Table 4. It can be seen that concerning the larger input graph size, a larger number of clusters $K$ help the model learn and obtain better performance.

Figure 4: Ablation studies of our model designs and the impact of $K$ (number of clusters) on model's performance.

| Model | TSP50 | | TSP100 | |
|---|---|---|---|---|
| | Obj. | Gap | Obj. | Gap |
| *w/o* multiresolution | 5.74 | 0.70% | 7.94 | 2.26% |
| *w/o* equivariant | 5.71 | 0.18% | 7.87 | 1.42% |
| Ours | 5.70 | 0.00% | 7.86 | 1.29% |
| Ours ($K = 5$) | 5.74 | 0.7% | 8.20 | 5.67% |
| Ours ($K = 10$) | 5.76 | 1.05% | 8.01 | 3.22% |

To further validate the faster convergence of our model via multiresolution graph training, we show additional results in Appendix D. Figure 6 indicates that leveraging multiresolution learning can help the model convergence faster and more stable.

## 6 CONCLUSION

The routing problems, e.g., TSPs and VRPs, are key topics in operations research due to their wide applications in transportation and logistics. In this work, we propose a novel multiple-resolution equivariant architecture that learns and extracts features effectively to generate better routing solutions. Moreover, we demonstrate empirically that our model is equivariant to input transformations and generalizes well to various input sizes, which significantly reduces the training costs in real-world applications and can have positive social impacts. Our research suggests that imposing symmetries and exploiting the multiscale structures are necessary for a wide range of other applications in operation research and optimization. In future work, we aim to apply our model to other NP-hard combinatorial optimization problems and develop a more general model that can find optimal solutions with minimal training and inference costs.

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

## A    MORE ON GROUP AND REPRESENTATION THEORY

**Definition A.1** (Group). A set $G$ with a binary operation $\circ : G \times G \to G$ is called a group if it satisfies the following properties:

- Closure: $\forall g, g' \in G, g \circ g' \in G$.
- Associativity: $(g \circ g') \circ g'' = g \circ (g' \circ g'')$ for any $g, g', g'' \in G$.
- Identity: $\exists e \in G$ such that $e \circ g = g \circ e = g$ for all $g \in G$.
- Inverse: $\forall g \in G$, there is an element $g^{-1} \in G$ such that $g \circ g^{-1} = g^{-1} \circ g = e$.

A group $G$ is finite if the number of elements in $G$ is finite. A group $G$ is called Abelian or commutative group if $g \circ g' = g' \circ g$ for every pair of $g, g' \in G$.

In order to build symmetry-preserving models, we are interested in how groups act on data. We assume that there is a domain $\Omega$ underlying our data. A group action of $G$ on a set $\Omega$ is defined as a mapping $(g, x) \mapsto g \cdot x$ associating a group element $g \in G$ and a point $x \in \Omega$ with some other point on $\Omega$ in a way that is compatible with the group operations, i.e.:

- $x = g \circ (g^{-1} \cdot x) = g^{-1} \circ (g \cdot x)$ for all $g \in G$ and $x \in \Omega$;
- $g \circ (h \cdot x) = (g \circ h) \cdot x$ for all $g, h \in G$ and $x \in \Omega$;
- If $G$ is Abelian then $g \circ (h \cdot x) = h \circ (g \cdot x)$ for all $g, h \in G$ and $x \in \Omega$.

Symmetry of the domain $\Omega$ imposes structure on the function $f$ defined on the domain. We define *invariant* and *equivariant* functions in Def. A.2 and Def. A.3, respectively.

**Definition A.2** (Invariance). A function $f : \Omega \to \mathcal{X}$ is $G$-invariant if $f(g \cdot x) = f(x)$ for every $g \in G$, i.e. regardless of how we transform the input by a group action, the function's outcome in some space $\mathcal{X}$ is unchanged.

**Definition A.3** (Equivariance). A function $f : \Omega \to \Omega$ is $G$-equivariant if $f(g \cdot x) = g \cdot f(x)$ for every $g \in G$, i.e. the group action on the input affects the output in a completely predictable way[2].

The most important kind of group actions is *linear* group actions, also known as *group representation* or simply *representation*. A linear action $g$ on signals on the domain $\Omega$ is understood in the sense that $g \cdot (\alpha x + \beta x') = \alpha(g \cdot x) + \beta(g \cdot x')$ for any scalars $\alpha, \beta \in \mathbb{C}$ and signals $x, x' \in \Omega$. We can describe linear actions either as maps $(g, x) \mapsto g \cdot x$ that are linear in $x$, or equivalently, as a map $\rho : G \to \mathbb{C}^{d \times d}$ that assigns to each group element $g \in G$ a $d \times d$ invertible matrix $\rho(g)$, satisfying the condition $\rho(g \circ h) = \rho(g)\rho(h)$ for all $g, h \in G$. Here, the matrix representing a composite group element $g \circ h$, i.e. $\rho(g \circ h)$, is equal to the matrix product of the representation of $g$ and $h$, i.e. $\rho(g)\rho(h)$.

Let $M_d$ be the set $d \times d$ complex matrices. A *matrix group* is a set of matrices in $M_d$ which satisfy the properties of a group under matrix multiplication. Formally, a *representation* $\rho$ of a group $G$ is defined as a function which maps $G$ to a matrix group, preserving group multiplication (see Def. A.4).

**Definition A.4** (Representation). Given a group $G$, a $d$-dimensional representation of $G$ is a matrix valued function $\rho : G \to \mathbb{C}^{d \times d}$ such that

$$\rho(g)\rho(h) = \rho(g \circ h),$$

for all $g$ and $h$ in $G$.

A *group homomorphism* is a map between groups that preserves the group operation (see Def. A.5). Alternatively, we can also define the representation based on homomorphism as in Def. A.6.

**Definition A.5** (Homomorphism). Let $G$ and $H$ be two groups and let $\circ$ denote the group operation. A homomorphism $\phi : G \to H$ is a map satisfying

$$\phi(g \circ g') = \phi(g) \circ \phi(g'),$$

for all $g$ and $g'$ in $G$.

**Definition A.6** (Representation). Let $G$ be a group and $V$ be a vector space over a field $\mathbb{F}$. A representation of $G$ over $V$ is a homomorphism

$$\rho : G \to \mathrm{GL}(V),$$

where $\mathrm{GL}(V)$ is the *general linear group* on $V$, or the group of invertible linear maps $\phi : V \to V$.

---

[2]Indeed, $G$-invariant is a special case of $G$-equivariant.

Symmetry of the signals' domain $\Omega$ imposes structure on the function $f$ defined on such signals. Given the help from representation, we can define *invariant* and *equivariant* functions more precisely, as in Def. A.7 and Def. A.8, respectively. Invariance means that regardless of how we transform the input by a linear group action (or multiplying with a group representation), the function's outcome in some space $\mathcal{X}$ is unchanged.

**Definition A.7** (Invariant fucntion). Given a group $G$ and its non-trivial representation $\rho$, a function $f : \Omega \to \mathcal{X}$ is $G$-invariant if

$$f(\rho(g)x) = f(x),$$

for all $g \in G$ and $x \in \Omega$.

**Definition A.8** (Equivariant fucntion). Given a group $G$, its non-trivial representation $\rho$ and another representation $\rho'$ (indeed, $\rho$ and $\rho'$ can be identical), a function $f : \Omega \to \Omega'$ is $G$-equivariant or $G$-covariant if

$$f(\rho(g)x) = \rho'(g)f(x),$$

for all $g \in G$ and $x \in \Omega$.

A neural network $\boldsymbol{f}$, in general, can be written as a composition of multiple functions $\{f_1, \ldots, f_L\}$ in which function $f_\ell$ is the $\ell$-th layer out of $L$ layers of the network:

$$\boldsymbol{f} = f_L \circ f_{L-1} \circ \ldots \circ f_1,$$

where $f_1$ is the bottom layer directly taking the input data and $f_L$ is the top or output layer making the decision. A $G$-equivariant neural network (or simply *equivariant neural network*) is the one composing of all $G$-equivariant layers. If the model outputs scalar values, the top layer $f_L$ is $G$-invariant. For example, Fig. 5 depicts Equivariant Graph Attention (Liao & Smidt, 2022) model that is equivariant with respect to permutation and rotation groups.

## A.1 Symmetricities in COs

Symmetricities are found in various COPs Kwon et al. (2020); Kim et al. (2022). We conjecture that imposing those symmetricities on $F_\theta$ improves the generalization and sample efficiency of $F_\theta$. We define the two identified symmetricities that are commonly found in various COPs:

**Definition A.9** (**Problem Symmetricity**). Problem $\boldsymbol{P}^i$ and $\boldsymbol{P}^j$ are problem symmetric ($\boldsymbol{P}^i \overset{\text{sym}}{\longleftrightarrow} \boldsymbol{P}^j$) if their optimal solution sets are identical.

**Definition A.10** (**Solution Symmetricity**). Two solutions $\boldsymbol{\pi}^i$ and $\boldsymbol{\pi}^j$ are solution symmetric ($\boldsymbol{\pi}^i \overset{\text{sym}}{\longleftrightarrow} \boldsymbol{\pi}^j$) on problem $\boldsymbol{P}$ if $R(\boldsymbol{\pi}^i; \boldsymbol{P}) = R(\boldsymbol{\pi}^j; \boldsymbol{P})$.

Several problem geometric symmetricity found in various COPs is the rotation, translation, and reflection:

**Theorem A.11** (**Geometric symmetricity**). *For any orthogonal matrix $Q$, the problem $\boldsymbol{P}$ and $Q(\boldsymbol{P}) \triangleq \{\{Qx_i\}_{i=1}^N, \boldsymbol{f}\}$ are problem symmetric: i.e., $\boldsymbol{P} \overset{\text{sym}}{\longleftrightarrow} Q(\boldsymbol{P})$.*

*Proof.* We prove the Theorem A.11, which states a problem $\boldsymbol{P}$ and its' orthogonal transformed problem $Q(\boldsymbol{P}) = \{\{Qx_i\}_{i=1}^N, \boldsymbol{f}\}$ have identical optimal solutions if $Q$ is orthogonal matrix: $QQ^T = Q^TQ = I$.

As we mentioned in A.1, reward $R$ is a function of $\boldsymbol{a}_{1:T}$ (solution sequences), $\|x_i - x_j\|_{i,j \in \{1,\ldots N\}}$ (relative distances) and $\boldsymbol{f}$ (nodes features).

For simple notation, let denote $\|x_i - x_j\|_{i,j \in \{1,\ldots N\}}$ as $\|x_i - x_j\|$. And Let $R^*(\boldsymbol{P})$ is optimal value of problem $\boldsymbol{P}$: i.e.

$$R^*(\boldsymbol{P}) = R(\boldsymbol{\pi}^*; \boldsymbol{P}) = R\left(\boldsymbol{\pi}^*; \{\|x_i - x_j\|, \boldsymbol{f}\}\right)$$

Where $\pi^*$ is an optimal solution of problem $\boldsymbol{P}$. Then the optimal value of transformed problem $Q(\boldsymbol{P})$, $R^*(Q(\boldsymbol{P}))$ is invariant:

$$
\begin{aligned}
R^*(Q(\boldsymbol{P})) &= R(\boldsymbol{\pi}^*; Q(\boldsymbol{P})) \\
&= R\left(\boldsymbol{\pi}^*; \{\|Qx_i - Qx_j\|, \boldsymbol{f}\}\right) \\
&= R\left(\boldsymbol{\pi}^*; \{\sqrt{(Qx_i - Qx_j)^T(Qx_i - Qx_j)}, \boldsymbol{f}\}\right) \\
&= R\left(\boldsymbol{\pi}^*; \{\sqrt{(x_i - x_j)^T Q^T Q(x_i - x_j)}, \boldsymbol{f}\}\right) \\
&= R\left(\boldsymbol{\pi}^*; \{\sqrt{(x_i - x_j)^T I(x_i - x_j)}, \boldsymbol{f}\}\right) \\
&= R\left(\boldsymbol{\pi}^*; \{\|x_i - x_j\|, \boldsymbol{f}\}\right) = R(\boldsymbol{\pi}^*; \boldsymbol{P}) = R^*(\boldsymbol{P})
\end{aligned}
$$

Therefore, the problem transformation of an orthogonal matrix $Q$ does not change the optimal value.

Then, the remaining proof is to show $Q(\boldsymbol{P})$ has an identical solution set with $\boldsymbol{P}$.

Let optimal solution set $\Pi^*(P) = \{\boldsymbol{\pi}^i(\boldsymbol{P})\}_{i=1}^M$, where $\boldsymbol{\pi}^i(\boldsymbol{P})$ indicates optimal solution of $\boldsymbol{P}$ and $M$ is the number of heterogeneous optimal solution.

For any $\boldsymbol{\pi}^i(Q(\boldsymbol{P})) \in \Pi^*(Q(\boldsymbol{P}))$, they have same optimal value with $\boldsymbol{P}$:
$$
R(\boldsymbol{\pi}^i(Q(\boldsymbol{P})); Q(\boldsymbol{P})) = R^*(Q(\boldsymbol{P})) = R^*(\boldsymbol{P})
$$

Thus, $\boldsymbol{\pi}^i(Q(\boldsymbol{P})) \in \Pi^*(P)$.

Conversely, For any $\boldsymbol{\pi}^i(\boldsymbol{P}) \in \Pi^*(\boldsymbol{P})$, they have sample optimal value with $Q(\boldsymbol{P})$:

$$
R(\boldsymbol{\pi}^i(\boldsymbol{P}); \boldsymbol{P}) = R^*(\boldsymbol{P}) = R^*(Q(\boldsymbol{P}))
$$

Thus, $\boldsymbol{\pi}^i(\boldsymbol{P}) \in \Pi^*(Q(\boldsymbol{P}))$.

$$
\text{Therefore, } \Pi^*(\boldsymbol{P}) = \Pi^*(Q(\boldsymbol{P})), \text{ i.e., } \boldsymbol{P} \xleftrightarrow{\text{sym}} Q(\boldsymbol{P}).
$$

## B  METHODOLOGY

**Motivation.**    Previous works on solving TSP problems for large input sizes usually simplify the original problem and find solutions for the simplified versions first, then combine them to obtain the final solution  (Li et al., 2021; Cheng et al., 2023; Kwon et al., 2020). The most common approach is divide and conquer, which splits the original graph into multiple smaller graphs and solves them separately. Then, it combines the local optima to generate the global one. However, this approach tends to get stuck in local optima because it is hard to design a good merging algorithm that can efficiently combine local paths. Motivated by this limitation, we propose an architecture that can transform the original complex graph into a simpler one without losing important features. After finding a solution for the simpler version, we use this information to help the original graph learn easier. Along with this, we also provide a symmetry-preserving graph attention network, which can learn equivariant solutions.

**Encoder.**    Figure 5 describes the details of our Equivariant Graph Attention Encoder. Our encoder consists of a linear layer and several equivariant graph attention layers ($N$ layers), each with four sublayers. Inspired by Transformer (Vaswani et al., 2017) and AM (Kool et al., 2019), each graph attention layer consists of two sublayers: Equivariant Graph Attention (EGA) layer and Feed-forward (FF) layer following the designs of Liao & Smidt (2022). In our setting, we use 8 attention heads with a dimensionality of 16. A feed-forward sublayer in each graph attention layer has a dimension of 512. We also use these settings for both TSP and CVRP.

**Training via Reinforce with greedy rollout baseline.**    We use the Rollout baseline proposed by Kool et al. (2019) as baseline $b(s)$ to train our model. We also compare the performance of our model to AM with the same Rollout baseline to demonstrate that the overall performance improvement is due to our multiresolution graph training strategy.

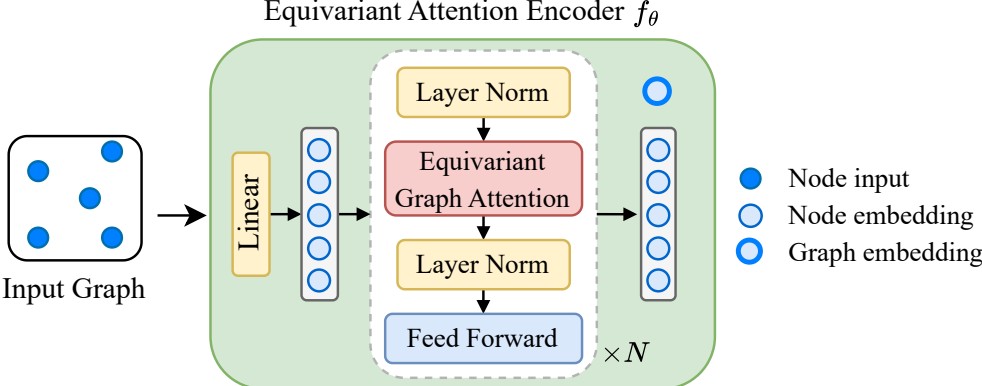

Figure 5: Equivariant Graph Attention based Encoder. Input nodes are embedded and processed by $N$ sequential layers, each consisting of a Layer Norm, Equivariant Graph Attention, and node-wise Feed-Forward sublayer. The graph embedding is computed as the mean of node embeddings. Best viewed in color.

## C  EXPERIMENTAL SETTINGS

**Training and hyperparameters.** For fair comparisons, the training hyperparameters are set the same as POMO (Kwon et al., 2020). Our model was trained for 2000 epochs (1 week) with a batch size of 256 on the generated dataset on the fly. We use 3 layers in the encoder with 8 heads and set the hidden dimension to $D = 128$. The optimizer is Adam with an initialized learning rate as $\eta = 10^{-4}$ and will be reduced every epoch with a learning rate decay is $1.0$. We implement our models in Pytorch (Paszke et al., 2019) and train all models on a machine with server A100 40GPUs. All the remaining parameters, such as random seed, are set the same as in Kwon et al. (2020). For clustering, we perform the K-means algorithm on PyTorch and execute it in parallel on GPUs to speed up training time and select 2D coordinates of the cities as features. The training time with the various problem sizes ranges from hours to 1 week. For the baselines, we use the same hyperparameters similar to the original works Kool et al. (2019); Kwon et al. (2020) that give the best performance for each problem, i.e. TSP and CVRP.

## D  ADDITIONAL RESULTS

### D.1  TRAVELLING SALESPERSON PROBLEM

**Scalability large-scale dataset.** We further investigate the scalability performance of our model by scaling our trained model on TSP100 datasets to larger instances on the TSPLIB benchmark with 50 to 400 nodes and 2D Euclidean distances (34 instances). The full results are shown in Table 3.

**Cross-distribution generalization.** For cluster distribution, we first sample 2D coordinates of $n_c$ cluster centroids and then sample cities around the centroids. For the mixed distribution, we sample $50\%$ of the cities from a uniform distribution and the rest from around cluster centers. For each distribution, we generate a test dataset of instances with a number of cities $n \in \{20, 50, 100\}$ and a number of clusters $n_c \in \{3, 5, 10\}$. Our model is trained on uniform datasets and tested on these datasets of the same size. More results of our model on TSP50 and TSP100 on the cluster and mixed distributions of datasets are shown in Table 4. The obtained results confirm that our model performs better AM for both distributions in all settings, especially in mixed distributions.

**Convergence performance analysis.** We validate the training performance of our model through learning curves compared to the baseline AM (Kool et al., 2019) and the optimal solution obtained by Concorde (Applegate et al., 2006). Figure 6 illustrates the learning curves of the models on TSP20 and TSP50. It can be observed that our model training is more stable and has faster convergence

Table 3: Generalization performance of our model compared with baseline AM on various real-world TSPs instances in TSPLIB. We report the average results across 10 independent runs per problem instance. The better results are in bold.

| Model | | POMO | | Sym-NCO | | Ours | |
|---|---|---|---|---|---|---|---|
| Instance | OTP | Obj | Gap (%) | Obj | Gap (%) | Obj | Gap (%) |
| eil51 | 426 | 429 | 0.70 | 432 | 1.41 | 426 | 0.00 |
| berlin52 | 7542 | 7545 | 0.04 | 7544 | 0.03 | 7544 | 0.03 |
| st70 | 675 | 677 | 0.30 | 677 | 0.30 | 679 | 0.59 |
| rat99 | 1211 | 1270 | 4.87 | 1261 | 4.13 | 1233 | 1.82 |
| kroA100 | 21282 | 21486 | 0.96 | 21397 | 0.54 | 21397 | 0.54 |
| kroB100 | 22141 | 22285 | 0.65 | 22378 | 1.07 | 22285 | 0.65 |
| kroC100 | 20749 | 20755 | 0.03 | 20930 | 0.87 | 20755 | 0.03 |
| kroD100 | 21294 | 21488 | 0.91 | 21696 | 1.89 | 21488 | 0.91 |
| kroE100 | 22068 | 22196 | 0.58 | 22313 | 1.11 | 22196 | 0.58 |
| eil101 | 629 | 641 | 1.91 | 641 | 1.91 | 641 | 1.91 |
| lin105 | 14379 | 14690 | 2.16 | 14558 | 1.24 | 14442 | 0.44 |
| pr107 | 44303 | 47853 | 8.01 | 47853 | 8.01 | 46572 | 5.12 |
| pr124 | 59030 | 59353 | 0.55 | 59202 | 0.29 | 59202 | 0.29 |
| bier127 | 118282 | 125331 | 5.96 | 122664 | 3.70 | 122664 | 3.70 |
| ch130 | 6110 | 6112 | 0.03 | 6118 | 0.13 | 6112 | 0.03 |
| pr136 | 96772 | 97481 | 0.73 | 97579 | 0.83 | 97481 | 0.73 |
| pr144 | 58537 | 59197 | 1.13 | 58930 | 0.67 | 58930 | 0.67 |
| kroA150 | 26524 | 26833 | 1.16 | 26865 | 1.29 | 26822 | 1.12 |
| kroB150 | 26130 | 26596 | 1.78 | 26648 | 1.98 | 26502 | 1.42 |
| pr152 | 73682 | 74372 | 0.94 | 75292 | 2.19 | 74372 | 0.94 |
| u159 | 42080 | 42567 | 1.16 | 42602 | 1.24 | 42567 | 1.16 |
| rat195 | 2323 | 2546 | 9.60 | 2502 | 7.71 | 2403 | 3.44 |
| kroA200 | 29368 | 29937 | 1.94 | 29816 | 1.53 | 29752 | 1.31 |
| ts225 | 126643 | 131811 | 4.08 | 127742 | 0.87 | 127742 | 0.87 |
| pr226 | 80369 | 82428 | 2.56 | 82337 | 2.45 | 82428 | 2.56 |
| Average | | | 2.11 % | | 1.90 % | | **1.23 %** |

Table 4: Predicted tour length of our models for TSP50 and TSP100 on clustered and mixed distributions.

| | Setting | | $n_c = 3$ | $n_c = 5$ | $n_c = 10$ |
|---|---|---|---|---|---|
| TSP50 | Cluster | POMO | 3.67 | 3.65 | 4.23 |
| | | Ours | 3.66 | 3.63 | 4.22 |
| | Mixed | POMO | 5.34 | 5.26 | 5.57 |
| | | Ours | 5.26 | 5.19 | 5.56 |
| TSP100 | Cluster | POMO | 4.97 | 5.08 | 5.68 |
| | | Ours | 4.94 | 4.90 | 5.54 |
| | Mixed | POMO | 7.02 | 7.04 | 7.35 |
| | | Ours | 6.92 | 6.89 | 7.22 |

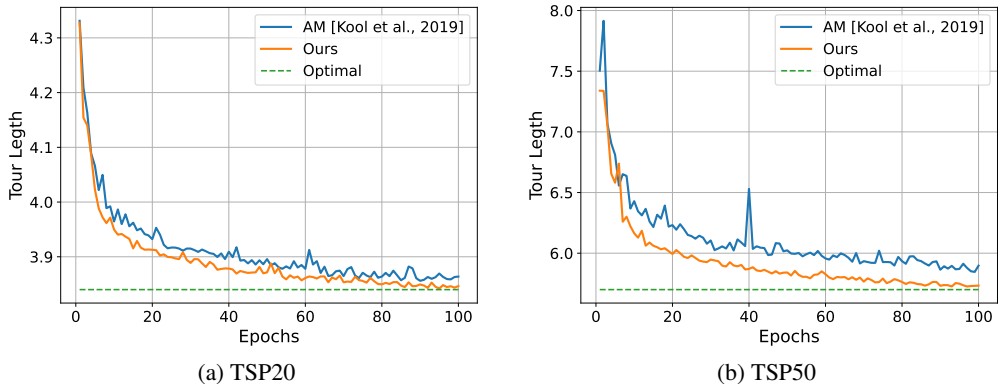

(a) TSP20            (b) TSP50

Figure 6: Learning curves of our model trained on 100,000 samples generated on the fly for TSP20 and TSP50. We compare our model with the baseline AM with the same settings to validate the convergence performance. The results are evaluated on a validation set of 10,000 random instances on the fly.

compared to AM with the same settings, thanks to its ability to parallelize the learning process across multiple resolutions (low-level and high-level) of the input graph.

## D.2 CAPACITATED VEHICLE ROUTING PROBLEM

We further assess the performance of our model for another well-known routing problem, e.g., Capacitated Vehicle Routing Problem (CVRP). The CVRP can be defined as follows. We have a depot node $0$ and a set of $n$ cities node $\{v_1, \ldots, v_n\}$, each city node $v$ has a demand $d_v$ to fulfill. There is a vehicle with capacity $C$ that need to start and end at the depot and visit a route of cities such that the sum of city demands along the route does not exceed $C$, i.e. $\sum_{v \in R_j} d_v \leq C$, where $R_j$ is $j$-th route.

Similar to (Kool et al., 2019), we generate instances with $n = 20, 50, 100$ cities, and normalize the demands by the capacities, i.e. $\hat{d}_{v_i} = \frac{d_{v_i}}{C}$, $i \in [\![1, n]\!]$. The demand $d_{v_i}$ sampled uniformly from $\{1, 2, ..., 9\}$ and $C = 30, 40, 50$ for $n = 20, 50, 100$, respectively.

## D.3 VISUALIZATION

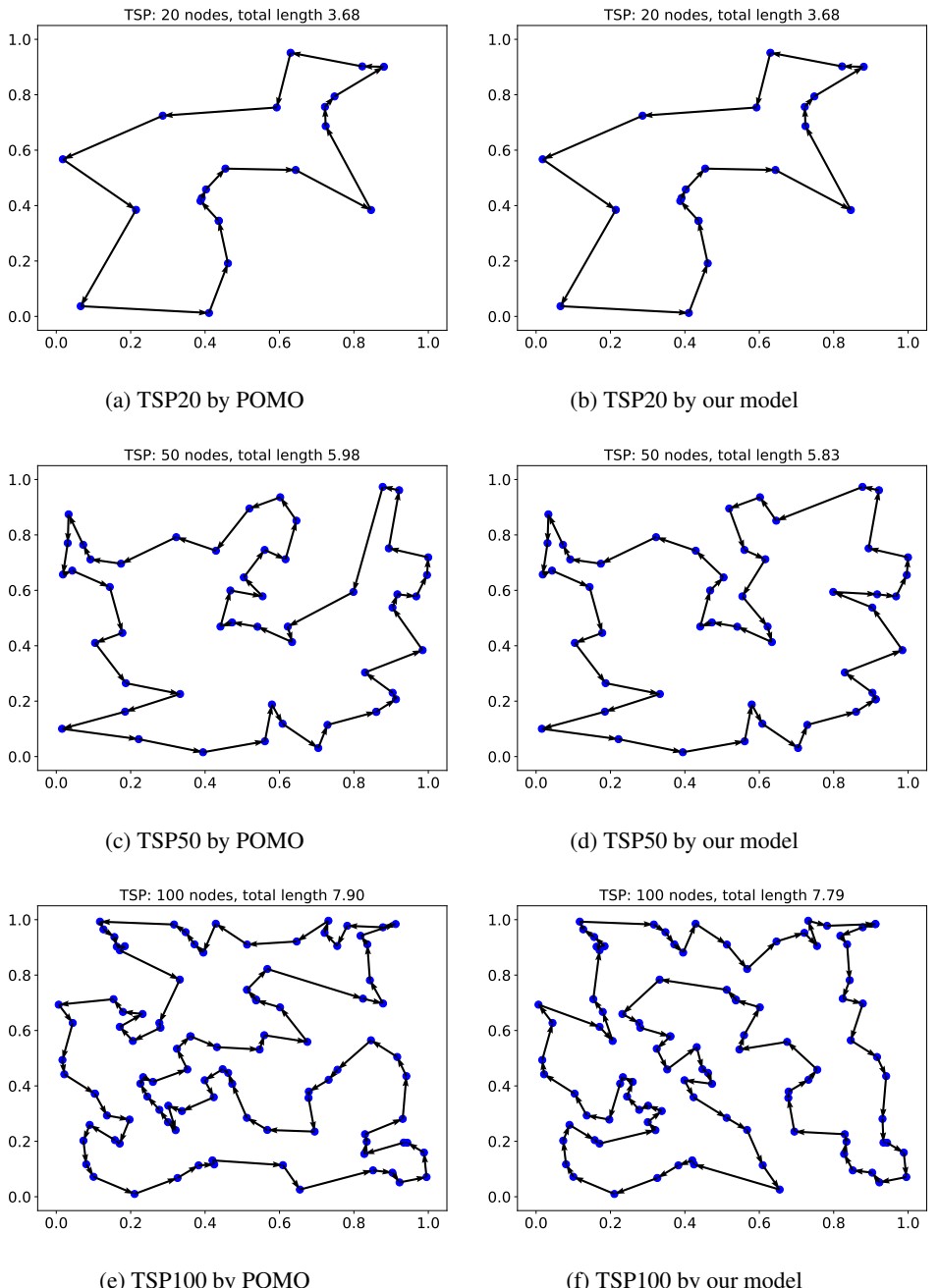

Figure 7: Example results on TSP20, TSP50, and TSP100 obtained by our model and the baseline AM trained on the same size.

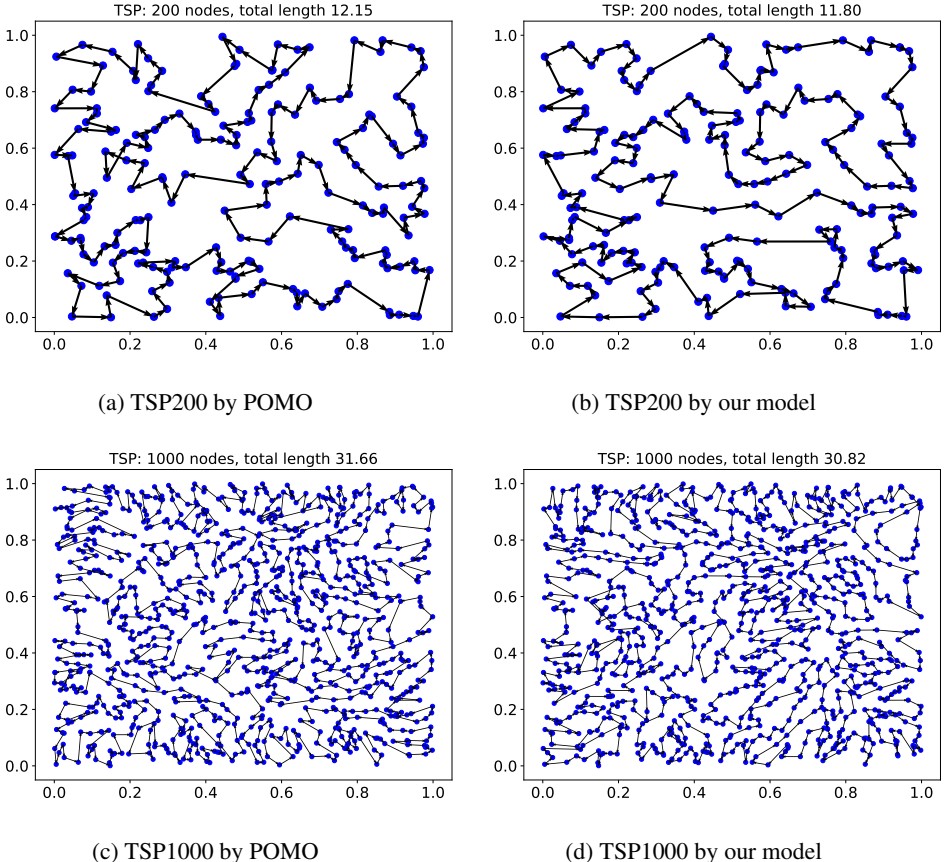

Figure 8: Example results on larger size TSP instances obtained by our model and the baseline AM trained on TSP100.

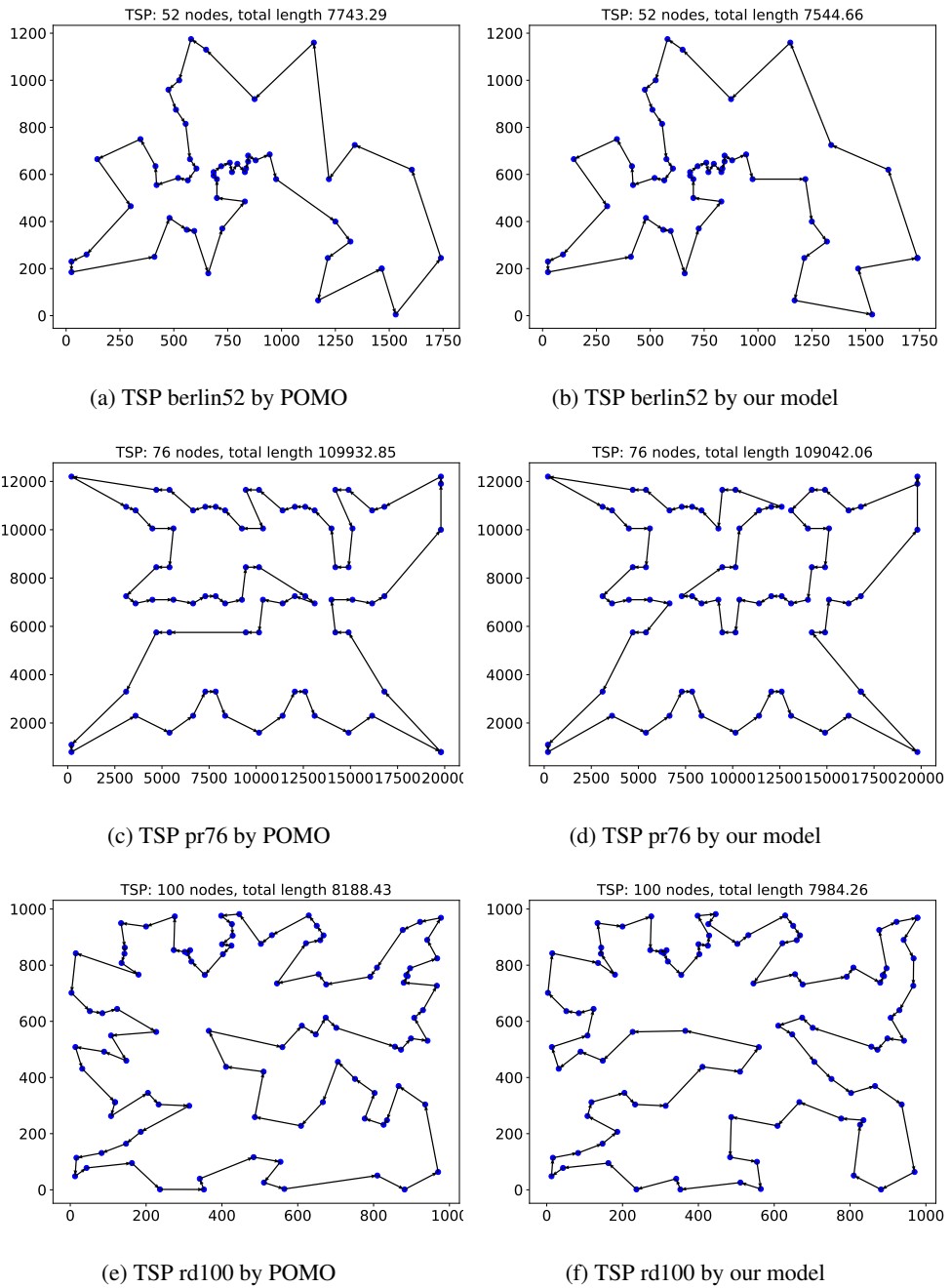

Figure 9: Example results obtained by our model and the baseline AM on real-world TSP instances in TSPLib.

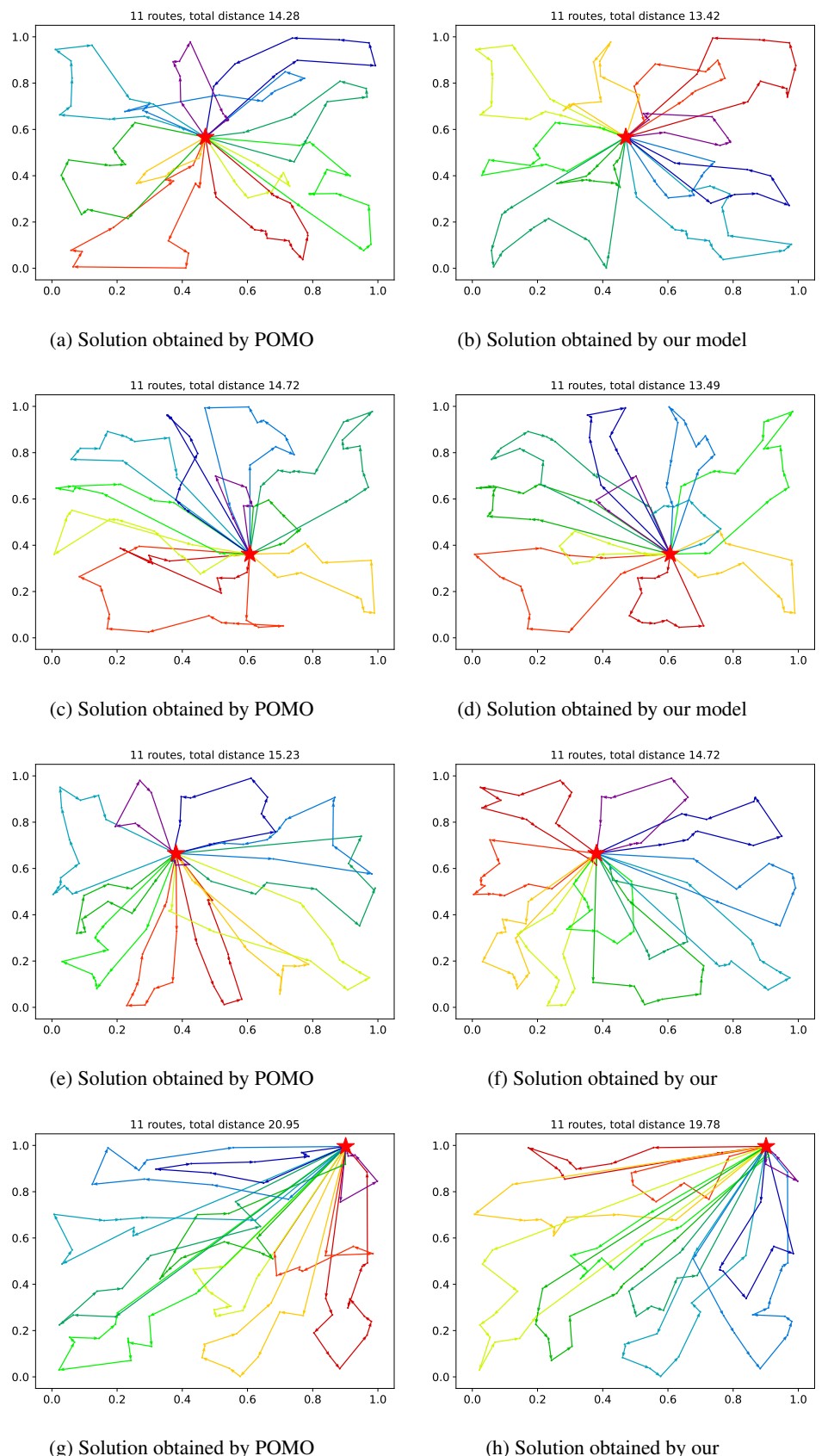

Figure 10: Example results on CVRP100 instances. The solutions obtained by our model and the baseline POMO trained on CVRP100.

