# OpenReview forum: "Symmetry-preserving graph attention network to solve routing problems at multiple resolutions"
_ICLR.cc/2024/Conference — ICLR 2024 Conference Withdrawn Submission_

### Official Review · Reviewer_JHgM · 2023-10-15

**Soundness:** 3 good
**Presentation:** 3 good
**Contribution:** 2 fair
**Rating:** 3
**Confidence:** 5

**Summary:**

In this paper, the authors introduce mEGAT to solve VRPs including TSP and CVRP, which leverages the symmetry property inherent to those problems. Overall, the mEGAT features two components, 1) the Equivariant Graph Attention network, which harnesses the problem symmetry operations such as rotation, translation, and scaling; 2) training at multiple level graphs such as the combination of sub-graph, high-level graph and the original graph. The experiments are conducted based on synthetic and benchmark instances, which exhibit better results of mEGAT than the selected learning-oriented baselines.

**Strengths:**

1- The overall presentation and writing are good;
2- The experiments are conducted on synthetic and benchmark instances.

**Weaknesses:**

1- While the idea of leveraging symmetry seems interesting, it is not novel. In fact, POMO and Sym-NCO also exploit them, but in different ways;
2- The baselines such as AM, POMO, AMDM, Sym-NCO are classic, but not the SOTA. Actually, quite a number of works on learning to solve VRPs  have been published in ICML, ICLR, AAAI this year and last year. Especially, many of them also test their performance on the benchmark instances. Without comparing with them, I am not full convinced by the results and conclusions drawn in this paper;
3- The studied problem sizes are quite small;
4- Overall, this work contributes little to the community of learning to solve VRPs.

**Questions:**

Please refer to the weakness.

---

### Official Review · Reviewer_sqmm · 2023-10-17

**Soundness:** 3 good
**Presentation:** 3 good
**Contribution:** 2 fair
**Rating:** 5
**Confidence:** 4

**Summary:**

This paper proposes the Multiresolution Equivariant Graph Attention Network (mEGAT) to solve vehicle routing problems (VRPs), which improves upon attention-based models [1, 2] (1) by completely respecting the symmetries of VRPs, including permutation, rotation, translation, and scaling, and (2) by considering both local and global information during training. With multiresolution graph training, the proposed method achieves superior performance on small-scale TSP and CVRP instances.

**Strengths:**

* The motivation is clear. Specifically, motivated by the divide and conquer, the authors propose to incorporate both local and global information (i.e., low-level and high-level graphs) into the training process. Motivated by the transformation invariance property, the authors use the equivariant graph attention network in the encoder layer. Overall, the proposed method is sound to me.
* The studied graph symmetry is interesting and valuable to the COP community.
* Experiments are conducted on both randomly generated and benchmark instances on two different problems.

**Weaknesses:**

* The related work section is not sufficient. A lot of literature on deep learning approaches for vehicle routing is not mentioned. Despite the cited papers, many of which were published before 2021, the authors should cite the latest literature. Given the fast development of the research on vehicle routing with deep learning, the related work is so simplified to me.

* The training problem size is too small for the current literature. You are using A100 GPUs, so why not train on larger problems? It raises concerns about the training efficiency and scalability of the proposed method. Could you report the training complexity of mEGAT compared with baselines?

* In Table 1, the authors directly copy the results from Sym-NCO [3]. Are their training settings the same as yours? If different, please retrain it following their open-source code. Moreover, the results of Sym-NCO on 20 and 50, and that of HGS [4] should be reported.

* For the variable-size generalization, the authors should test on larger sizes (e.g., TSP100+). It is expected that your model (trained on TSP100) performs well on TSP20 and TSP50 due to the (multiresolution) training on low-level graphs. But what if the problem sizes scale up?

* The authors should carefully proofread the submission since some minor issues exist:
  * On page 6, "$\pi_{\ell}^{sub}$ denotes the $\ell$-th resolution's solution on the high-level instance" -> $\pi_{\ell}^{high}$
  * In Algorithm 2, "Multisolution graph Training" -> "Multiresolution"
  * The caption of Table 3: "AM" -> "POMO". The best result is not in bold.

[1] Attention, learn to solve routing problems! In ICLR 2019.
[2] POMO: Policy optimization with multiple optima for reinforcement learning. In NeurIPS 2020.
[3] Sym-NCO: Leveraging Symmetricity for Neural Combinatorial Optimization. In NeurIPS 2022.
[4] https://github.com/vidalt/HGS-CVRP
[5] http://vrp.galgos.inf.puc-rio.br/index.php

----

**Given all,** I vote for rejection. This submission needs a bit more work on the empirical evaluation before meeting the bar of ICLR.

**Questions:**

* Any results on CVRPLIB [5]?

---

### Official Review · Reviewer_aD1V · 2023-10-25

**Soundness:** 3 good
**Presentation:** 4 excellent
**Contribution:** 2 fair
**Rating:** 3
**Confidence:** 4

**Summary:**

In this paper, the authors present a neural method for the solution of routing problems. Compared to previous efforts, two main innovations are introduced: first, an equivariant encoder GNN is used, in order to make the resulting representation invariant by permutations, rotations, and translations; secondly, a multi-scale training framework is devised, in which the model is trained jointly on a routing problems as well as sub-problems of it obtained via clustering of the associated graph. The authors claim and demonstrate that these enhancements result in better performance (as measured via tour length, evaluation time, and optimality gap) compared to existing neural baselines.

**Strengths:**

- The paper clearly written and the subject matter is well-presented.
- The methods used are well-described.
- The authors compare their approach against an exhaustive set of baselines, which they re-implemented themselves in some cases.
- The method proposed does beat all those baselines.

**Weaknesses:**

- The method does perform better than baselines, but the gap is not that extensive.
- Handcrafted solvers still remain clearly superior to all neural approaches, meaning that any real impact or the field of Operations Research is still only potential.
- Relatively small (up to n=100 cities) problem instances are considered, in line with previous ML efforts. Handcrafted solvers can comfortably handle problem instances several orders of magnitude larger.
- When it comes to the multi-scale training component of the method, it is not proven beyond doubt that the observed performance improvements are due to the leveraging of multi-scale information as the authors suggest; the improvement could just be the consequence of effectively training on more problem instances (the original one, and the sub-instances, and the high-level instances) compared to not using multi-scale training.
- Usually, equivariance (or any inductive bias in general) is mostly useful for improving data efficiency. Since TSP instances can be easily generated synthetically (which the authors themselves do), it does not appear that this use-case is one in which data are scarce, and therefore data-efficiency is an important consideration.

**Questions:**

- In order to verify their claim on the usefulness of multi-scale training, the authors should compare with a baseline (or even just a version of their method with the multi-scale scheme ablated away) which has been trained on a number of problem instances equivalent to the total number of instances and sub-instances that their main method is trained on. This would be an apples-to-apples comparison which would clarify that the leveraging of multi-scale information is indeed the cause of their method's success, as opposed to just effectively training on more data
- In general, can the authors clearly state over how many problem instances they train their method, and the baselines?
- Can the authors elaborate on why they do not also make their method scale-invariant, despite the fact that they correctly point out to scale invariance as being a property of TSP solutions?
- Could the authors motivate (via the citation of relevant work) their statement that previous learning approaches mostly learn from "local information"?
- In section 4.1, the authors state that "Many CO problems exhibit spatial locality"; while this is true for routing problems, it would be appropriate to be more specific and provide some examples. There are many CO problems without any notion of locality (or space) at all.
- Can the authors clarify what their "Evaluation time" metric refers to? Does it include the time for training? And how can it be different for e.g. Kool et al. if the same decoder is used? Since the authors' main architectural contribution is at the encoder level, why would they expect the inference time (assuming that this is what they report) to be impacted by it and be a metric of interest?

**Details Of Ethics Concerns:**

No concerns.

---

### Official Review · Reviewer_UYvA · 2023-10-30

**Soundness:** 3 good
**Presentation:** 3 good
**Contribution:** 2 fair
**Rating:** 3
**Confidence:** 3

**Summary:**

A deep learning model is proposed based on symmetries to solve routing problems. Symmetric instances extend training data and different sizes of instances are covered in the training. Symmetric graph attention network is used as encoder to retain symmetries of representations in DRL. The model is experimentally better than some literature.

**Strengths:**

Symmetry issue is critical in graph representation learning and routing problems if they are seen as graphs. The good use of symmetry benefits the sound information embedding. Claimed by authors, multiresolution of a routing problem benefits better policy learning if the neural network is powerful enough to learn a comprehensive policy from different sizes.

**Weaknesses:**

A lot of paradigms on deep learning for routing problems are in literature. But the paper only covers much less and neglects recent ones. Even the highly related ones are not included. As an example, Revisiting Transformation Invariant Geometric Deep Learning: Are Initial Representations All You Need solves TSP by symmetry-preserving approaches.

The compared methods are not new and most published before 2021. Therefore the results are not convincing. Moreover, the compared methods are supposed to be trained by multiple sizes of instance for fairness. The tested sizes are relatively small and no hints are described to raise the generalization scales.

**Questions:**

The work includes much more hyperparameters to tune such as L and K. How these hyperparameters are fast tuned given a routing problem need to be explained.

The ablation study showcases w/o equivariant has a slight impact on the performance. Therefore it is important to know how much more training time with the equivariant instances is traded to get the little improvement.

The description "current learning-based methods often focus on learning from local information (spatial locality)" confuses me. However,  Attention Model encodes relations between all nodes, which already covers the global information in it. Therefore the viewpoint in the description needs to be justified.